# Rapid meniscus-guided printing of stable semi-solid-state liquid metal microgranular-particle for soft electronics

Gun-Hee Lee [1,2,7], Ye Rim Lee[1,7], Hanul Kim [3], Do A Kwon [1], Hyeonji Kim[1], Congqi Yang[4], Siyoung Q. Choi [3,5], Seongjun Park [4,6], Jae-Woong Jeong [2,6 ✉] & Steve Park [1,6 ✉]

Liquid metal is being regarded as a promising material for soft electronics owing to its distinct combination of high electrical conductivity comparable to that of metals and exceptional deformability derived from its liquid state. However, the applicability of liquid metal is still limited due to the difficulty in simultaneously achieving its mechanical stability and initial conductivity. Furthermore, reliable and rapid patterning of stable liquid metal directly on various soft substrates at high-resolution remains a formidable challenge. In this work, meniscus-guided printing of ink containing polyelectrolyte-attached liquid metal microgranular-particle in an aqueous solvent to generate semi-solid-state liquid metal is presented. Liquid metal microgranular-particle printed in the evaporative regime is mechanically stable, initially conductive, and patternable down to 50 μm on various substrates. Demonstrations of the ultrastretchable (~500% strain) electrical circuit, customized e-skin, and zero-waste ECG sensor validate the simplicity, versatility, and reliability of this manufacturing strategy, enabling broad utility in the development of advanced soft electronics.

[1] Department of Materials Science and Engineering, Korea Advanced Institute of Science and Technology (KAIST), 291 Daehak-ro, Yuseong-gu, Daejeon 34141, Republic of Korea. [2] School of Electrical Engineering, Korea Advanced Institute of Science and Technology (KAIST), 291 Daehak-ro, Yuseong-gu, Daejeon 34141, Republic of Korea. [3] Department of Chemical and Biomolecular Engineering, Korea Advanced Institute of Science and Technology (KAIST), 291 Daehak-ro, Yuseong-gu, Daejeon 34141, Republic of Korea. [4] Department of Bio and Brain Engineering, Korea Advanced Institute of Science and Technology (KAIST), 291 Daehak-ro, Yuseong-gu, Daejeon 34141, Republic of Korea. [5] KAIST Institute for the NanoCentury, Korea Advanced Institute of Science and Technology (KAIST), Daejeon 34141, Republic of Korea. [6] KAIST Institute for Health Science and Technology, 291 Daehak-ro, Yuseong-gu, Daejeon 34141, Republic of Korea. [7] These authors contributed equally: Gun-Hee Lee, Ye Rim Lee. ✉email: jjeong1@kaist.ac.kr; stevepark@kaist.ac.kr

With the rising demand for electronic devices for applicability in wearable displays, electronic skin (e-skin), and wearable healthcare devices, customized soft electronics with high stretchability are receiving a great deal of attention[1–6]. Unlike conventional strategies for patterning using rigid masks (e.g., screen printing, photolithography), digital mask-based additive patterning such as ink-jet printing allows rapid and cost-effective fabrication of electronic devices of various designs[7–10]. Furthermore, since rigid-mask-based patterning is not compatible with soft substrates, additive printing has been considered more appropriate for the versatile fabrication of soft electronics.

An ideal conductive material and printing process for soft stretchable electronics simultaneously requires (1) high conductivity, (2) high-resolution, (3) high stretchability, (4) mechanical stability, (5) simple one-step rapid processing, and (6) printability on various substrates. As candidates, 1/2D materials, conducting polymers, and metal particle-embedded matrix have been investigated[11–15]. However, these materials have limitations for use as soft stretchable electronics, due to their low conductivity compared to metals and/or insufficient mechanical deformability. Gallium-based liquid metal (LM) overcomes these limitations owing to its unique combinations of metal-level high conductivity and deformability[16–18]. However, LM's mechanical instability, due to its fluidity, hinders its practical applicability, as direct contact with other materials (e.g. electronics components, skin) are limited[19].

To overcome the abovementioned drawbacks of bulk LM, LM particle-based approach has been actively studied where the outer oxide layer can restrain its fluid-like behavior[20,21]. However, the formation of the native oxide layer deteriorates the electrical conductivity since the oxide is an insulator. Hence, after patterning, additional processing such as mechanical scrubbing, tensile straining, or chemical etching is required to rupture or remove the oxide layer[10,22–25]. This, however, converts the LM particles back to bulk LM, thus reintroducing the issues faced with bulk LM. Furthermore, the need for additional processing potentially introduces defective (e.g. open or shorted) areas, thus generating uncertainties that make the fabrication process unreliable. To cope with this matter, Jeong et al. recently reported intrinsically conductive liquid metal microparticles by doping with hydrogen, which can be patterned by nozzle printing[26]. Although the doped-liquid metal particles show reliable intrinsic electrical conductivity while maintaining their stability, high-resolution patterning was not demonstrated. Furthermore, the organic solvent-based ink and the need to anneal at high temperatures for an extended period of time (120 °C, 3 h) restrict the type of substrate that can be used and limit rapid manufacturing. Therefore, despite the numerous efforts, the abovementioned critical requirements for printed conductive material for soft stretchable electronics have not yet been addressed.

Herein, we introduce meniscus-guided printing (MGP) of semi-solid-state polyelectrolyte-attached liquid metal microgranular-particles (PaLMP) to pattern highly stable, ultra-stretchable, and initially conductive electrodes with high-resolution in a simple one-step process. Figure 1a depicts the MGP process of PaLMP. The term 'meniscus' refers to the curved liquid-air interface that naturally forms between the laterally moving nozzle and the substrate. Solvent evaporation occurring at the meniscus facilitates thin-film deposition as the nozzle moves. Figure 1b–h pictorially demonstrates various features of PaLMP-based printed electrodes. The printed electrodes are initially conductive (Fig. 1b), and can be printed at resolution down to 50 μm (Fig. 1c). Rapid multi-layered patterning is possible with a one-step printing session as no additional post-

processing for each layer is required. Large-area patterning (Fig. 1d), ultra-high stretchability (Fig. 1e), and mechanical stability (Fig. 1f) can be utilized to integrate our electrodes with conventional electronics such as micro-LED displays (Fig. 1g). Since our technique does not contain harsh processing (use of organic solvents, annealing at high temperature), our electrodes can be printed on various substrates (e.g. VHB tape, Polydimethylsiloxane (PDMS), hydrogel, biogel, metal film, wafer, glass, PET) (Fig. 1h), making our technique highly versatile for the fabrication of various types of soft electronic devices. Supplementary Table 1 presents a comparison between the conventional LM particle-based soft electronics fabrication and our MGP-based strategy.

## Results

**Preparation and characterization of ink for MGP.** Our ink was formulated by tip-sonicating a solution consisting of LM and polystyrene sulfonate (PSS, molecular weight (MW): 1,000,000 g/mol) in water containing 10 vol.% acetic acid (AA). Tip sonication generates LM particles with a thin oxide shell, as depicted on the left side of Fig. 2a and Supplementary Fig. 1[20,27]. Without PSS, the zeta potential (ζ) of the LM particles was +76.3 mV (inset of Fig. 2b), indicating that the LM particles are positively charged in the solution. With the inclusion of PSS, ζ is −23.2 mV (inset of Fig. 2b), suggesting that the PSS are surrounding the LM particles through electrostatic interaction (right side of Fig. 2a, Supplementary Fig. 2); these LM particles surrounded with PSS will be referred to as PaLMP in this work. LM particles without PSS will be referred to as LMP.

Figure 2b shows the UV-vis spectra of various solutions (all solutions had the same solvent consisting of water and AA): solvent only, LM, PSS, and LM + PSS. The LM + PSS solution had a spectrum representative of the combination of LM and PSS spectra. Supplementary Fig. 3 shows the Energy dispersive X-ray spectroscopy (EDS) mapping, indicating that PSS are well-attached on the surface of LM particles. Even after adding a strong acid (HCl) to our solution, which typically eliminates the oxide shell and converts the LM particles into bulk LM droplet[28], the PaLMP remained well-dispersed (Supplementary Fig. 4), which furthermore corroborates the surrounding of LM particles with PSS to enhance the chemical stability of PaLMP.

The well-dispersed and enhanced stability of PaLMP was critical to reliably print the electrodes, as presented in Fig. 2c. The nozzle extruding LMP ink was often clogged during printing (Supplementary Movie 1); on the other hand, the PaLMP ink showed reliable printing without clogging (Supplementary Movie 2). Clogging can be explained by the coalescence of the LMP during printing due to its instability (Supplementary Fig. 5). Computational fluid dynamics was carried out using finite element method (FEM), as seen in Fig. 2d. 'Printing pressure' was defined as the pressure required to print the ink at the same speed of 6 mm/s. As the radius of the droplet ($R_{drop} > 40$ μm) becomes similar to that of the radius of the nozzle ($R_{nozzle} = 50$ μm), the printing pressure increases drastically (square data points) beyond the typical working range (≤250 Pa), indicating that nozzle clogging will occur. Detailed mathematics and geometry are given in Supplementary Figs. 6 and 7, Table 2, and Note. Rheological property of the PaLMP ink is presented in Supplementary Fig. 8.

For meniscus-guided thin-film deposition, wettability of the ink is essential to reliably and compactly deposit the film onto the substrate[29,30]. Figure 2e is the contact angle of the LMP and PaLMP ink on glass, showing that the latter has lower contact angle. The PaLMP ink droplet was partially withdrawn using a pipette. The radius of the droplet remained constant, leading to

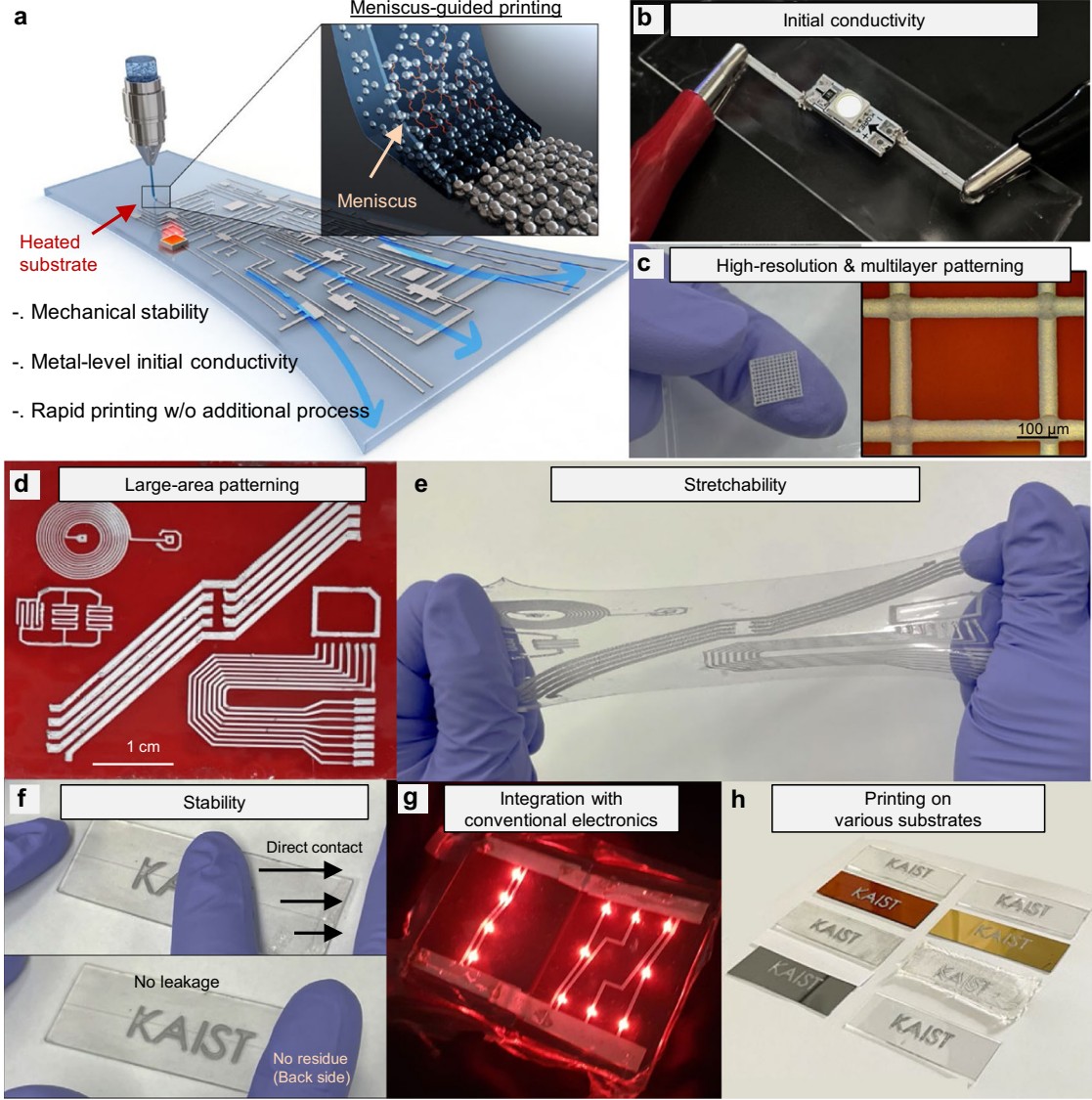

**Fig. 1 Meniscus-guided printing (MGP) process of polyelectrolyte-attached liquid metal microgranular-particles (PaLMP) and demonstrated characteristics. a** Schematic illustration of MGP process of PaLMP. **b–h**, Photograph of demonstrated characteristics: initial electrical conductivity (**b**), high-resolution/multi-layer patterning (**c**), large-area patterning (**d**), stretchability (**e**), mechanical stability (**f**), integration capability with conventional electronics (**g**), printing capability on various substrates (**h**).

further decrease in the contact angle, signifying that the droplet is pinned to the substrate. These results confirm that PSS improves the wettability of the ink. The better wettability of the PaLMP ink also lowers the contact angle and extrudes out the meniscus during printing as presented in Fig. 2f (see Supplementary Fig. 9 for real-time meniscus observation set up). These effects accelerate solvent evaporation due to reduced surface-to-volume ratio of the solution, enabling rapid drying and film formation simultaneously with the moving nozzle.

For compact assembly of PaLMPs during printing, which is required for high film conductivity, AA is essential. When PaLMPs are printed with deionized water (DI)-based solvent, voids are observed throughout the film; whereas, compactly assembled PaLMP is observed with AA in DI-based solvent (Fig. 2g). The improved packing density can be explained by the increase in proton concentration, which works as charge neutralizing agent to reduce the electrostatic repulsion between the negatively charged PaLMPs (Fig. 2h)[31]. The change in zeta potential value from −68.1 to −23.2 mV with the addition of AA

suggests the reduced electrostatic repulsion between PaLMPs (Supplementary Fig. 10).

**MGP for high-resolution, well-adhered, and initially conductive PaLMP film.** Figure 3a is a photograph of MGP of PaLMP. A pneumatically-driven printing head and a heated substrate at 70 °C were utilized for MGP to facilitate solvent evaporation and achieve robust adhesion between PaLMPs and substrate. Rather than ejecting droplets as in ink-jet printing, MGP pins the ink to the substrate at constant pressure and drags the nozzle across the substrate at a fixed vertical distance from the substrate (Fig. 3b). This generates a meniscus (e.g. concaved air-liquid interface), at which solvent evaporation and continuous film deposition occur[29]. The thickness of the film decreased with increasing printing speed (Fig. 3c), indicating that printing is being conducted in the evaporative regime[29,32] (Supplementary Fig. 11 is surface profile of printed PaLMP films with different thicknesses). In this regime, solvent evaporation and film formation occur concurrently with the moving nozzle, and this was

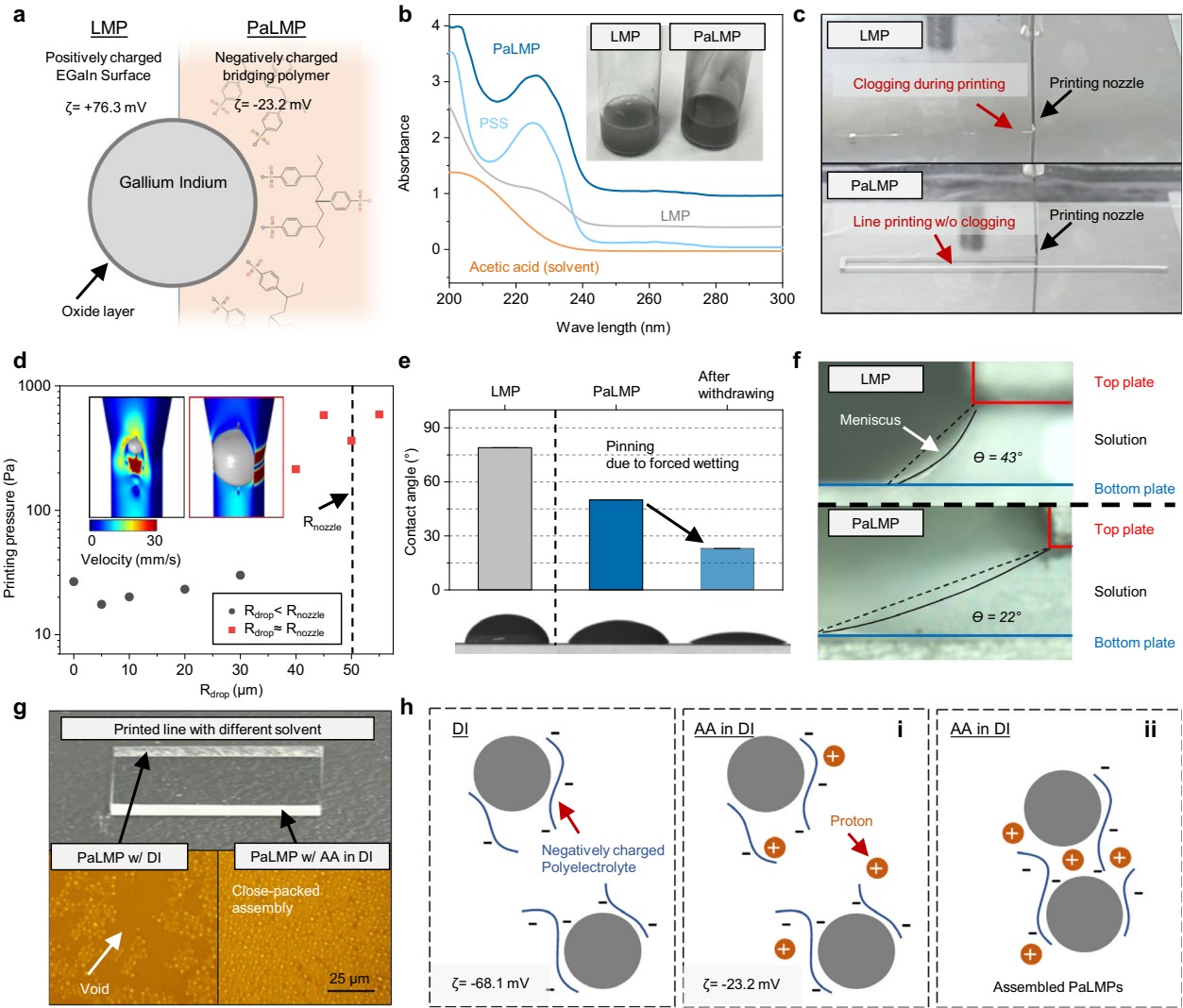

**Fig. 2 Chemical configuration and characterization of PaLMP-based ink. a** Schematic illustration and zeta potential of liquid metal particles (LMP) and PaLMP configuration. **b** UV-vis spectra of ink according to various combinations. Inset: Photograph of each ink **c** Photograph of MGP with LMP and PaLMP-based ink. LMP-based ink is easily clogged. **d** The printing pressure varying the size of liquid metal drop. Inset rainbow bar is linear from 0 to 30 mm/s. **e** Contact angles of LMP and PaLMP-based ink, and PaLMP-based ink after withdrawing half of the original volume. **f** OM side view image of the meniscus formed with LMP and PaLMP-based ink. **g** Photograph and OM image of printed PaLMP with different solvent. **h** Schematic illustration of PaLMPs assembly process in different solvent.

critical to obtain closely-packed particle assembled architecture robustly adhered on the surface without particle rupturing (Fig. 3d) and high-resolution features that match diameter of the nozzle opening (Supplementary Fig. 12). As seen in Fig. 3e, line width as small as 50 μm can be printed via MGP.

The peel-off test of the printed film on Si wafer reveals robustness of the adhered film (Fig. 3f for peel-off test on wafer substrates). On the contrary, a film coated with an ink absent of AA and a film printed without heating the substrate had relatively poor adhesion (Supplementary Fig. 13). pH of the ink containing AA decreased as solvent evaporated at 70 °C (Fig. 3g). Therefore, we project that during MGP, as the solvent evaporates away near the air-liquid-film boundary (i.e. contact line), the PaLMPs are annealed in an acidic condition as they assemble into a film. X-ray photoemission spectroscopy (XPS) reveals that when ink containing AA and heated substrate is used, gallium peaks are present along with the gallium oxide peaks. Without AA in the ink or without heating the substrate, only gallium oxide peaks were observed (Fig. 3h). Thus, the annealing of PaLMPs in an acidic condition partially removes the oxide shell, which extrudes

out the gallium. We hypothesize that the removal and reformation of the oxide layer at the interface strengthen the adhesion of PaLMP film to the substrate[21,33] (Supplementary Fig. 14). We have furthermore observed that only the film printed with an AA-containing ink on a heated substrate was initially conductive, with a conductivity of $1.5 \times 10^6$ S/m. This can be attributed to the partial removal of the oxide during chemical annealing, which electrically connects the PaLMP via partial merging. Not needing any additional activation step to make the electrode conductive is critically important as this complicates the fabrication process and potentially introduces defective regions (e.g. shorted or open circuit). Finally, PaLMP ink-based film was generated using screen printing; however, this film had a different film morphology and was not stretchable (further explanation can be found in Supplementary Figs. 15 and 16), which further indicates the importance of MGP.

**Stretchability of coated PaLMP film.** Figure 4 presents schematic and actual images of different films being stretched on

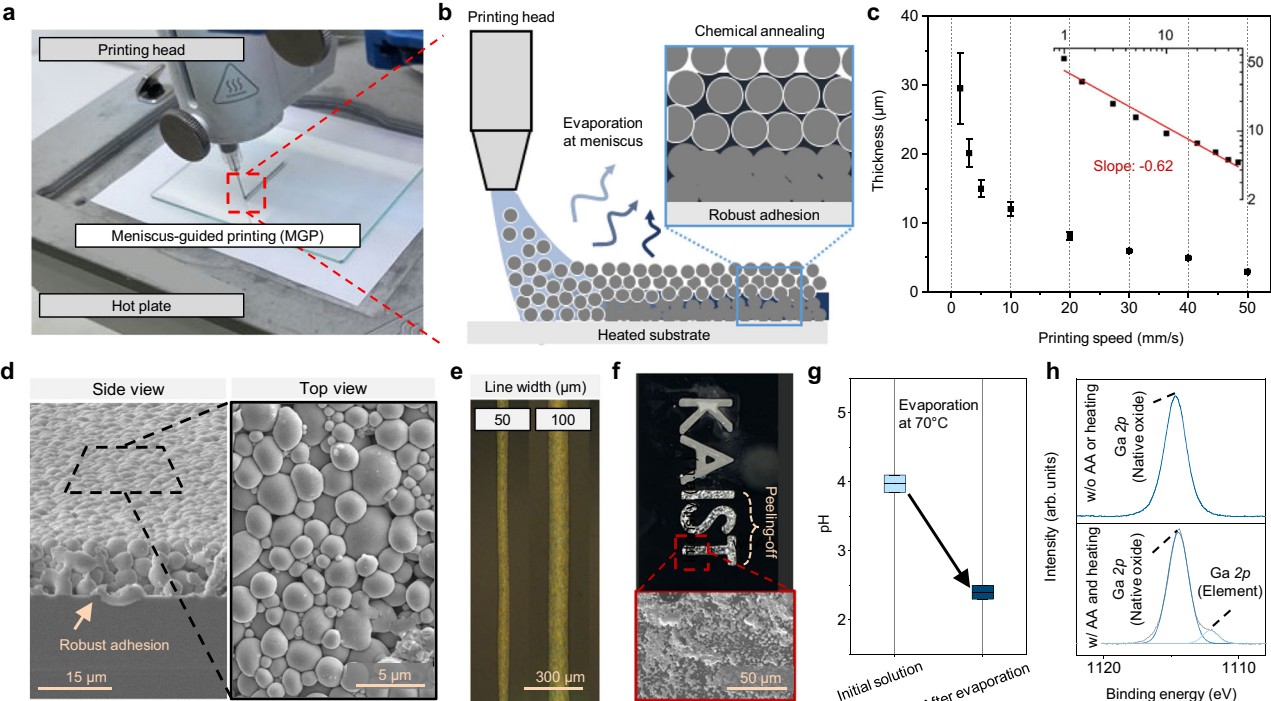

**Fig. 3 MGP of PaLMPs. a** Photograph image of MGP of PaLMPs. **b** Schematic illustration of MGP of PaLMP with acetic acid (AA)-based ink. **c** Thickness of deposited PaLMPs according to printing speed. Error bars indicate maximum and minimum values. **d** SEM images of printed PaLMP through MGP. **e** OM image of PaLMPs printed with different widths. **f** Photograph and SEM image of printed PaLMP after peeling off with scotch tape. **g** pH variation of designed ink before and after evaporation. Error bars indicate maximum and minimum values. **h** XPS spectra of printed PaLMP under different coating conditions.

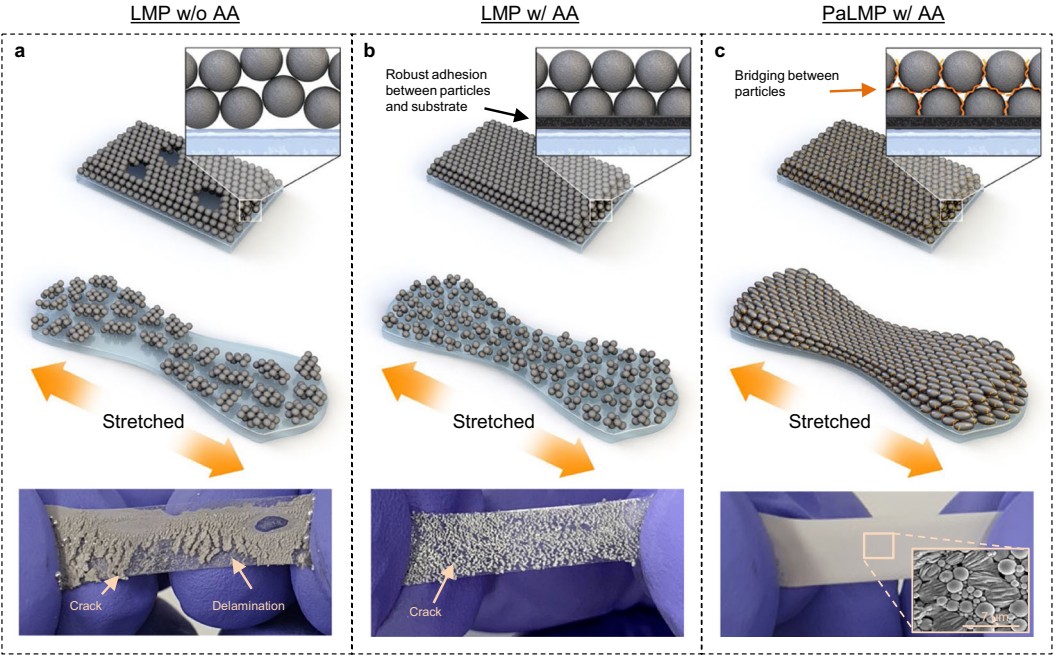

**Fig. 4 Stretchability of coated LMP according to combinations of additives. a** Schematic illustration and photograph of coated LMPs without AA under application of strain. Severe delamination and cracks are observed. **b** Schematic illustration and photograph of coated LMPs with AA under application of strain. **c** Schematic illustration, photograph, and SEM image of printed PaLMPs with AA under application of strain. There are no observed delamination or cracks.

PDMS substrate. The film prepared with an ink absent of both AA and PSS cracked and delaminated from the substrate under strain (Fig. 4a). For the film made with an AA-containing ink without PSS, it adhered well onto PDMS; however, the film cracked under strain (Fig. 4b). These results reconfirm the role of AA in enhancing the adhesion of LMP-based film onto the substrate. For the film generated with an ink containing both AA and PSS, the film neither cracked nor delaminated under strain (Fig. 4c). SEM image confirms elongation of particle without rupturing after strain (see Fig. 5c below for repeated strain cycle).

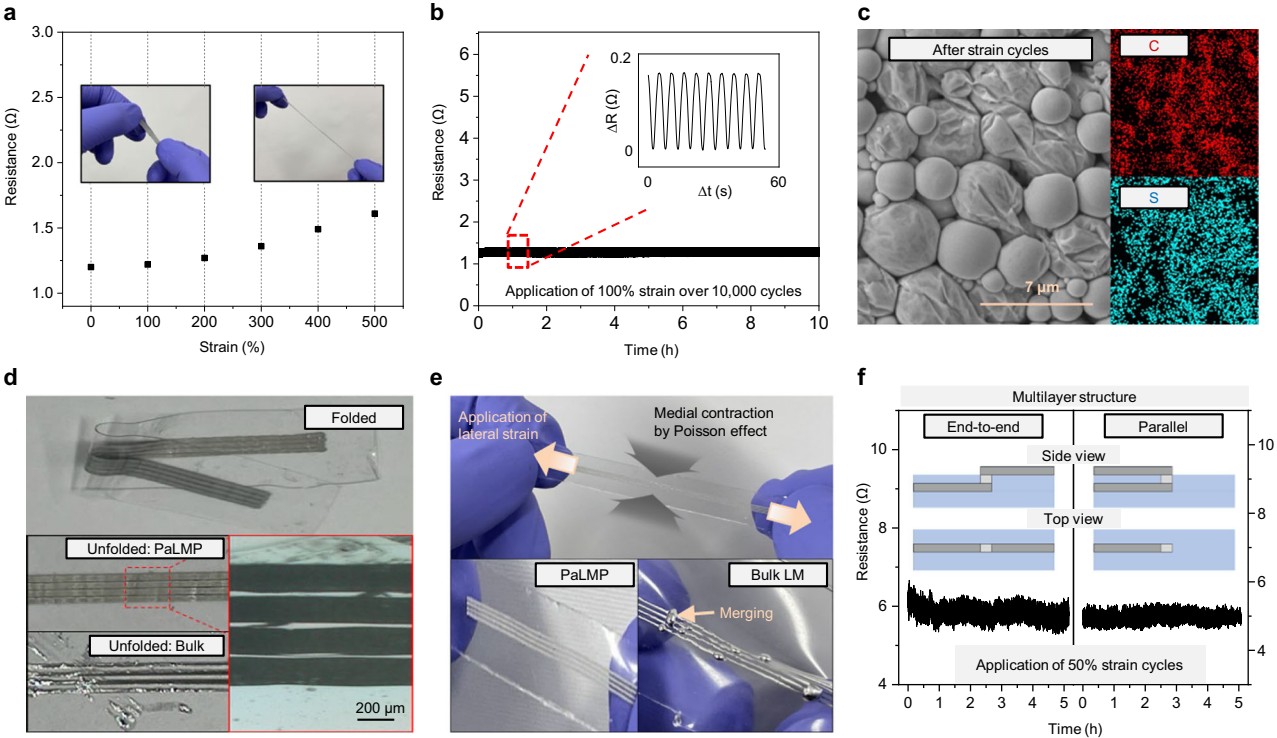

**Fig. 5 Characterization of PaLMP. a** Resistance of printed PaLMP according to applied strain. **b** Resistance of PaLMP under repeated application of 100% strain over 10,000 cycles. **c** SEM image and EDS mapping of PaLMP after the strain cycles. There is no rupturing in PaLMP, and carbon and surfer contained in PSS are uniformly distributed after the cycling. **d** Top: photograph of folded PaLMP. Middle left, bottom right: photograph and OM images after unfolding of folded PaLMP, respectively. Bottom left: photograph after unfolding of folded bulk LM. **e** Top: photograph of stretched PaLMP. Bottom: images of PaLMP (left) and bulk LM (right) after stretching. **f** Resistance variation of multilayered PaLMP-based interconnects under repeated application of 50% strain. Inset: illustration of end-to-end (left) and parallel (right) electrodes configuration.

This confirms that PSS not only encapsulates the liquid metal particles to prevent GaIn from being extruded out, but also bridges the particles together[34,35] to render the film stretchable. (see Supplementary Fig. 17 for the force as a function of the strain of PDMS coated with PaLMP).

**Electrical property and mechanical stability of PaLMPs.** Figure 5a shows the resistance change of PaLMP printed on a VHB tape according to strain. Unlike bulk LM that shows resistance variance according to the change of geometry[36,37], printed PaLMP film shows negligible resistance variation under strain (e.g. an increase of 0.1 Ω at 200% strain) due to its piezo positive conductivity[38,39]. Such a minor change in resistance with strain is likely due to the increasing number of electrical pathways caused by the tight bridging between PaLMPs under strain. Interestingly, this electro-mechanical property can be tuned by changing the thickness of printed PaLMPs. For example, thin PaLMP film shows a lower conductance and large variation of resistance according to strain (Supplementary Fig. 18), while thick PaLMPs offer negligible change in resistance when deformed. Such tunable nature of PaLMP makes it highly versatile for the design of various soft electronics – the thin film can be utilized in strain sensors and deformable heaters, and the thick film can be employed for stretchable electrodes and interconnects. The PaLMP film shows reliable and stable response under the application of repeated strain (over 10,000 cycles) without electrical failure (Fig. 5b). No rupturing of particles and no detachment of polyelectrolyte were found after the strain cycling, as presented in the SEM image and EDS mapping of carbon and sulfur (both of which are elements in PSS) (Fig. 5c). The

dependence of PSS molecular weight on PaLMP-based film properties is presented in Supplementary Fig. 19.

The fluidity of bulk LM often causes an unwanted electrical short or failure when the substrate is deformed[40]. Therefore, conventional LM-based electronics should be encapsulated with an additional insulating matrix. However, this design requirement hinders applicability and practical utilization of LM-based electronics by limiting their direct electrical contact with other electronic components. Unlike bulk LM, PaLMP exhibits excellent mechanical stability while maintaining its metallic electrical conductivity (Fig. 5d, e). When the substrate is folded, patterned bulk LM smears off due to its fluidity (bottom left photograph in Fig. 5d). Furthermore, due to its ultra-high surface tension, unwanted merging between patterned lines occurs, which causes electrical shorting. On the contrary, no flowing, rupturing or merging is found in the PaLMP-based patterned line after repeated folding-unfolding, as presented in the photograph (middle left) and OM image (bottom right) in Fig. 5d. Similar phenomenon can be observed when the devices with multiple close lines are stretched in the lateral direction. Stretching results in contraction in the perpendicular directions due to the Poisson effect (see Supplementary Fig. 20 for details), as shown in the top photograph in Fig. 5e. Contraction induced by stretching causes unwanted merging and electrical shorting of bulk-LM-based lines patterned in close proximity; thereby, limiting construction of high-density patterns required for compact electronics (bottom right of Fig. 5e). In contrast, PaLMP-based lines are mechanically stable (bottom left of Fig. 5e); therefore, electrical shorting is not of concern even under extreme straining. Furthermore, the enhanced stability of PaLMP allows direct integration of multilayer structures and conventional electronic components on it,

thus enabling facile fabrication of complex printed circuits[41]. For example, we successfully fabricated multilayered interconnects joined through VIA with end-to-end and parallel electrodes configurations. Repeated strain testing shown in Fig. 5f validates mechanical robustness and reliability of the multilayered interconnects, which are essential for high-density soft electronics. Also, PaLMP shows stable electrical conductivity without any encapsulation over several weeks (Supplementary Fig. 21). Finally, to confirm its integration capability with conventional electronics, we demonstrated the fabrication of a deformable micro-LED display (Supplementary Fig. 22 and Fig. 1g) by connecting the conventional electronic components with PaLMP devices using silver paste and conductive epoxy. Furthermore, electronic components can be directly integrated with PaLMP without conductive paste (Supplementary Fig. 23).

**Demonstration of customizable, scalable e-skin and deformable EMG sensor with PaLMP.** Initially conductive and stretchable PaLMP can be easily integrated on various substrates with simple MGP methods. The additive manufacturing approach of printing facilitates rapid fabrication of various electronics with different sizes and shapes[42]. This customizability and scalability along with direct printability on soft substrates make MGP of PaLMP highly appropriate for creation of target-optimized soft electronics such as e-skin. Figure 6a is a photograph of optimally customized e-skin attached to different sizes of prosthetic hands. Demonstrated e-skin consists of PaLMP-based multilayer pressure sensing array and strain sensors. The enhanced mechanical stability of PaLMP enables the multilayer structuring and the usage as a direct contact electrode for piezoresistive pressure sensing (Fig. 6b). The multilayered pressure

sensing array was designed for precise sensing of spatial pressure distribution without noise, which can be caused by electrical shorting between interconnects (see Supplementary Fig. 24 for overall fabrication process and Supplementary Fig. 25 for fabrication process of pressure sensor). The strain sensor for an artificial finger was built with a serpentine resistor and stretchable interconnects with negligible resistance change (Fig. 6c). For reliable strain sensing that does not get affected by stretching of interconnects, interconnects were printed thick while the serpentine resistor was printed thin by varying the printing speed (Supplementary Fig. 26). Since MGP allow rapid and customized deposition of PaLMPs without the need for additional process for electrical activation, various designs and dimensions of conductor patterns can be printed and used on-the-fly. This feature facilitates creation of e-skins optimized for different sizes or parts of robots and prosthetics. Both the pressure sensing array and the strain sensor fabricated by MGP showed reliable functionality, as demonstrated with sophisticated sensing of applied pressure distribution and finger bending, respectively (Fig. 6d, e, and Supplementary Figs. 27, 28).

The customizable manufacturability of the printing approach can also bring significant impact on the design of wearable electronics such as skin-mounted EMG sensors. To effectively monitor activity of target muscles with high accuracy, EMG sensors should be optimally designed for targeted body parts or for specific individuals with unique sizes and configurations of their skeleton and muscle[43]. Here, the versatile printing-based manufacturability and high mechanical/chemical stability of PaLMP enable construction of deformable EMG sensors customized for desired applications, as presented in Fig. 6f. Another significant attribute of the PaLMP-based EMG electrode

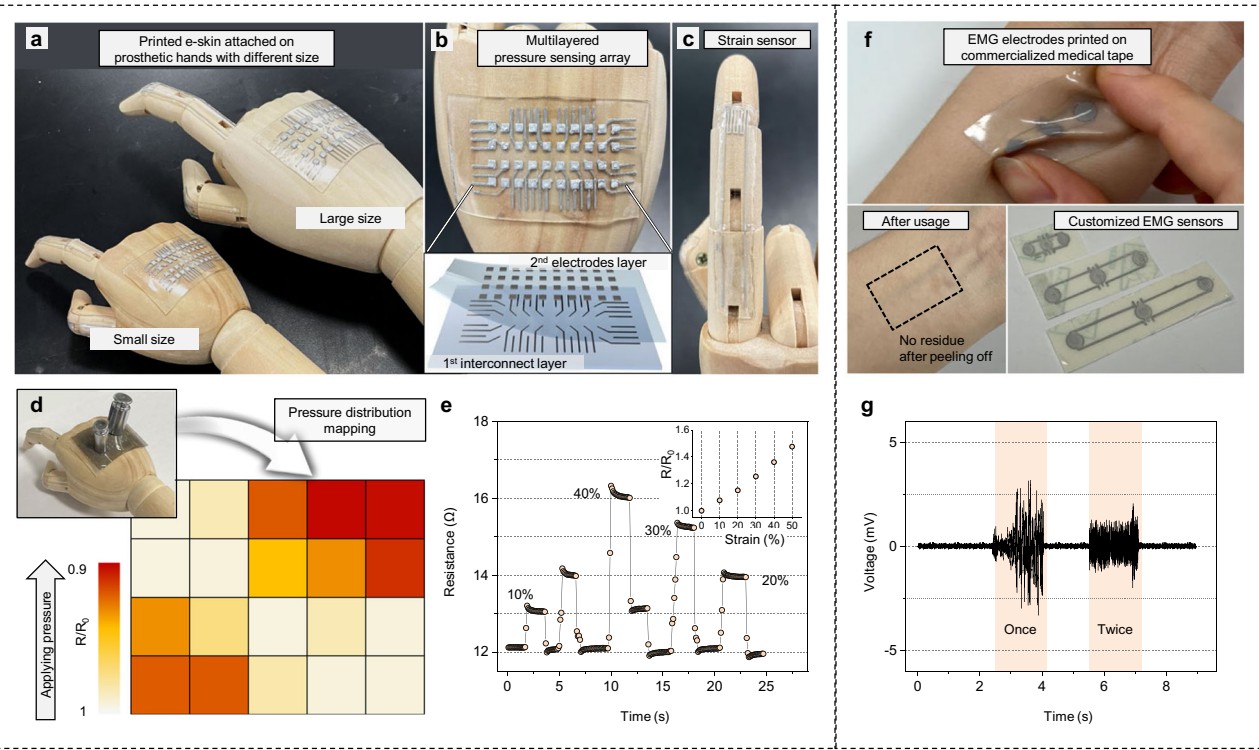

**Fig. 6 PaLMP-based customizable e-skin and deformable EMG sensor. a** Photograph of PaLMP-based customized e-skin attached to prosthetic hands with different sizes. **b** Photograph and illustration of multilayer pressure sensing array. **c** Photograph of strain sensor. **d** Pressure distribution mapping with multilayer pressure sensing array. **e** Real-time resistance monitoring with strain sensor. Inset: relative resistance according to strain (**f**) Photograph of deformable PaLMP-based EMG sensor fabricated on the commercialized medical tape. No residue is observable after detaching the EMG sensor from the skin. **g** EMG signals measured with PaLMP-based EMG sensor on forearm.

is that it is biocompatible (Supplementary Fig. 29) and does not leave residues on the skin after usage, making it highly suitable for skin-interfaced use. Figure 6g shows EMG signals measured with a PaLMP-based sensor on a forearm, verifying its ability of high fidelity EMG sensing (setup for physiological signal measurement is presented in Supplementary Fig. 30). These demonstrations together show customizability as well as reliability of MGP of PaLMP for facile and rapid establishment of soft electronics optimized for specific target applications.

**Demonstration of zero-waste ECG sensor with PaLMPs.** Many of medical devices including skin-mounted devices are generally used only once to avoid any potential infection issues caused by their re-use. However, with the increasing demand for the single-use healthcare devices[44], disposing of its waste becomes a serious economical and environmental problem[45,46]. To cope with this matter, zero-waste medical electronics with biodegradable materials have been gaining a great deal of attention[47–49]. LM has been drawn attention in this regard since it is recyclable through the reduction into bulk LM with acid. Based on PaLMP that is easily patternable on various substrates, we demonstrated its possibility for the functionalized soft zero-waste medical electronics.

Figure 7a shows a zero-waste skin-attachable ECG sensor robustly adhered on the wrist, which consists of PaLMPs electrode and sticky degradable biogel. Excellent wettability and stability of ink and immediate evaporation of solvent during MGP allow the reliable printing of PaLMPs on the dynamic surface of biogel. To monitor the ECG signals, three PaLMP-based electrodes are attached on the right arm (RA), left arm (LA), and left leg (LL) for real-time ECG monitoring as depicted in Fig. 7b. Measurements and comparisons of ECG signals acquired with PaLMP-based electrodes and commercialized electrodes (Fig. 7c) demonstrate the high fidelity of PaLMP-based electrodes for ECG monitoring.

Different from many skin-attachable sensors based on conventional metals[50–53], this PaLMP-based ECG sensor can be fully disposed without waste as presented in Fig. 7d. First, biodegradable substrate is fully degraded by immersing in the diluted weak acid solution (5% AA). Subsequently, all of PaLMPs were reduced by the addition of HCl (37%) and merged into bulk LM droplets that are reusable. This demonstration shows the possibility of using PaLMP for construction of skin-compatible, deformable, and recyclable electronics, suggesting new opportunities for zero-waste medical devices.

## Discussion

An advanced manufacturing strategy that enables rapidly customizable and scalable fabrication of soft electronics can open many opportunities for personal electronics, robotics, and medical devices by providing optimized designs for desired purposes in a swift manner[8,42,54,55]. Here, we developed such a technique based on MGP of PaLMP. PaLMP patterned with MGP is not only highly customizable in terms of design, but also stretchable, mechanically stable, and capable of high-resolution and multi-layer patterning on various substrates without any additional process. Printing of stable, stretchable, and initially conductive PaLMP overcomes the limitations of bulk LM and LMP-based approach[56–59], where their mechanical and chemical instability and the need for an additional process for electrical activation prevent facile construction of advanced soft, stretchable electronics. Demonstrations of highly stretchable yet mechanically stable interconnects, deformable display with conventional LED, customized multilayered e-skins, and zero-waste ECG sensors validate its applicability, reliability, and versatility for soft electronics. We envision that MGP of PaLMP can be important groundwork for advanced soft electronics for many real-world applications.

## Methods

**Materials**. All chemicals were used without further purification and were acquired from Sigma-Aldrich unless otherwise mentioned. To prepare and characterize the PaLMP ink, eutectic gallium indium alloy (EGaIn, Rich-Metals), poly(styrene sulfonate) (PSS, with an average molecular weight of 70,000 and 1,000,000), and acetic acid (99%) were used. For printing the substrate, PDMS (Sylgard 184, Dow Corning), VHB tape (3 M), Tegaderm (3 M), and bio-gel[47] were used. For the fabrication of pyramid structured pressure sensor, PDMS, trichloro(1H,1H,2H,2H-perfluorooctyl)silane (PFOCTS, 97%), and pyrrole monomer, polypyrrole solution, and Iron(III) p-toluenesulfonate hexahydrate were included.

**Preparation of PaLMP ink for MGP**. Bulk LM (1.25 g) and 0.07 g ($7 \times 10^{-8}$ mol) of PSS were added in diluted acetic acid (10 vol.% in DI water, 2 ml). This compound solution was tip sonicated (VC 505, Sonics & Materials, 3 mm microtip) at 500 W and 20 kHz for 30 min.

**MGP process of PaLMPs**. A commercial nozzle printer (BIO X6, CELLINK) was used for MGP of PaLMPs. Prepared ink (2 mL) was injected into the syringe for nozzle printing and the printing bed was heated to 40~70 °C to accelerate solvent evaporation and to facilitate chemical annealing during MGP. Nozzle types and printing speeds were determined according to the purpose (diameter: 50~200 μm, printing speed: 0.5~30 mm/s)

**Fabrication of multi-layer pressure sensing array**. Schematic illustration of the overall fabrication process of multi-layer pressure sensing array is depicted in the Supplementary Fig. 18. First, the interconnected lines were printed on the PDMS substrate (thickness: 300 μm). Subsequently, insulating layers were deposited on areas of the patterned substrate other than via hole. Top electrodes, in contact with

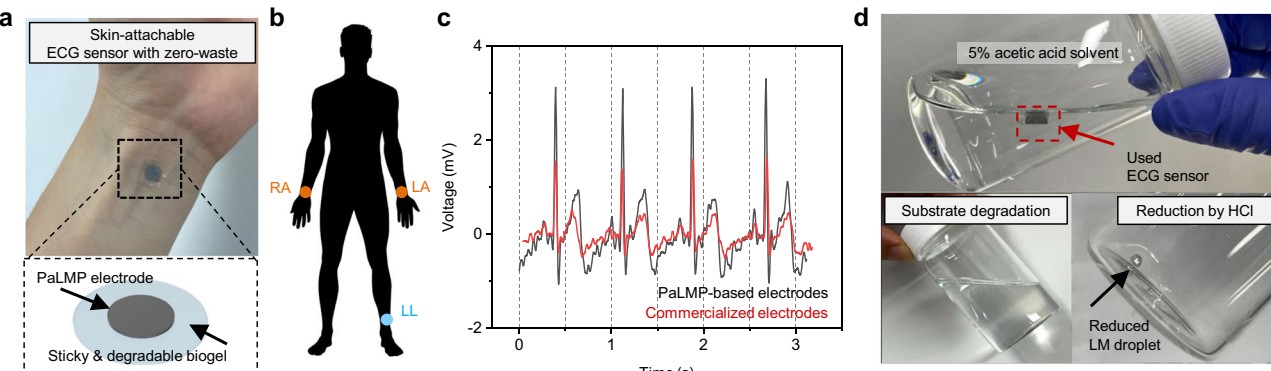

**Fig. 7 PaLMP-based zero-waste ECG sensor. a** Photograph image and illustration of skin-attachable ECG sensor with zero-waste. **b** Schematic illustration of three electrodes-based ECG monitoring. ECG sensors are attached on the right arm (RA), left arm (LA), and left leg (LL) **c** Real-time ECG monitoring with PaLMP-based electrodes and commercialized electrodes. **d** Photograph image of used ECG sensor disposal process. PaLMP-based electrodes are reduced into bulk LM and the biogel-based substrate is fully degraded.

the pressure sensor, were printed on the insulating layer by aligning with the hole for electrical connection with interconnected line.

**Fabrication of pressure sensor**. Schematic illustration of the overall fabrication process of the pressure sensor is depicted in Supplementary Fig. 19. Silicon substrate was etched in pyramidal shapes and subsequently, PFOCTS was chemically deposited on the etched substrate to facilitate the peel-off. Pre-cured PDMS was poured on the etched silicon and spin-coated. Molded PDMS was cured at 80 °C for 30 min, and was peeled off from the silicon. The pyramid-structured PDMS was treated with oxygen plasma and polypyrrole monomer was deposited on the surface. After monomer deposition, PDMS was immersed in the solution containing the catalyst (Iron(III) p-toluenesulfonate hexahydrate dissolved in water at 1.6 wt%) and polypyrrole[60].

**Fabrication of artificial finger**. Schematic illustration of the overall fabrication process of artificial finger is depicted in Supplementary Fig. 20. To maximize resistance change under strain, the serpentine structure of thin PaLMP (thickness: 3 μm) was printed for strain sensor, whereas thick PaLMP (thickness: 20 μm) was printed for interconnects.

**Fabrication of functional physiological signal sensor**. PaLMP-based bio-interfaced electrodes were printed on commercialized medical tape and bio-gel for EMG and ECG measurement, respectively without further treatment. Fabricated EMG and ECG sensors are connected with a commercialized Bluetooth EP signal sensing system (BioRadio, Great Lakes NeuroTech.).

**Characterization**. Chemical, rheological, and morphological characterization.

*Chemical characterization*. Zeta potential values of inks were measured by DLS (Zetasizer nano zs, Malvern). Each ink was characterized by UV/VIS Spectrophotometer (Lambda 1050, Perkin Elmer) under wavelengths ranging from 200 nm to 500 nm. To determine the chemical composition of the printed PaLMP under different coating conditions, X-Ray Photoelectron Spectroscopy (XPS, K-alpha, Thermo VG Scientific) was conducted.

*Rheological characterization*. The apparent viscosity of LM inks with different proportions of PSS was measured by MCR 302 rheometer (Anton Paar) at a shear rate of $10^{-2} \sim 10^{2}\,\mathrm{s}^{-1}$ at room temperature. To determine the wettability of each ink, the contact angle was measured with a contact angle analyzer (SEO Phoenix). Measurements were conducted twice each sample: when 100 μl of the sample was dropped and when 50 μl of that was withdrawn. To observe the morphology of the printed PaLMP, SEM images were taken by S4800 (Hitachi).

*Electrical characterization*. To measure the resistance of PaLMP characteristic, LCR meter (4284 A, HP) was used. The samples were printed and cut to the same size (printing: 0.3 mm × 20 mm, cutting: 1 mm × 25 mm) unless stated otherwise.

Characterization of the e-skin was conducted with the LCR meter, force gauge (the maximum force is 50 N, Mark-10), a stand with a motor (Mark-10), and a customized manual stain machine. Real-time monitoring of EMG and ECG was conducted with commercial wireless electrophysiology measurement equipment (BioRadio, Great Lakes NeuroTech.).

**Experiments on human subjects**. All experiments on human skins were performed under approval from the Institutional Review Board at Korea Advanced Institute of Science and Technology (protocol number: KH2021-231). All subjects voluntarily involved in experiments after informed consent.

## Data availability
The authors declare that the data supporting the findings of this study are available within the article and its Supplementary Information files. Extra data or source files are available from the corresponding author upon reasonable request.

## Code availability
The software code used for rheological modeling of ink is available from the corresponding author upon reasonable request.

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

## Acknowledgements
This work was supported by the National Research Foundation of Korea (NRF-2021M3H4A1A03049075 and NRF-2020M3C1B8A01111568).

## Author contributions
G.-H.L., Y.R.L., J.-W.J., and Steve P. conceived the concept and designed experiments. G.-H.L. designed the ink, performed the chemical and electrical characterization, and conducted data analysis. Y.R.L. conducted the experimental work, including ink preparation, nozzle printing, and soft electronics fabrication, and performed the mechanical characterization. Hanul K. conducted a computational simulation regarding the rheological behavior of ink. D.K. and Hyeonji K. assisted fabrication of soft electronics. C.Y. and Seongjun P. conducted a biocompatibility test. S.Q.C. provided comments regarding data analysis. J.-W.J. and Steve P. were responsible for managing all aspects of this project. G.-H.L. wrote the draft. J.-W.J. and Steve P. revised the manuscript. All authors discussed the results and the manuscript.

## Competing interests
The authors declare no competing interests.
