## [Peer Review File · Nature Communications]

Rapid meniscus-guided printing of stable semi-solid-state liquid metal microgranular-particle for soft electronicsREVIEWER COMMENTS

Reviewer #1 (Remarks to the Author):

Key Results: The manuscript describes the development of a liquid metal based ink that is suitable for meniscus-guided printing. The work includes control experiments to elucidate the effect of different ink components on the strain-dependent conductivity of the materials. A key part of the work is time-dependent pH as the aqueous solvent evaporates, which enables chemical dissolution of the oxide layer between microparticles, improving the conductivity. The ink is printed from an aqueous solvent, which is consistent with green processing concepts.

Validity: The manuscript includes some robust and well-supported conclusions, while a few of the conclusions are not well supported. These are discussed in the section on suggested improvements.

Significance: The work addresses several challenges with previous liquid metal printing methods that include limited print resolution and challenges with adhesion. Demonstrations include reprocessability of the starting materials through solvent dissolution, and piezoresistivity coefficients that depend on the thickness of the electrode. Consequently, this work includes significant technical insights into ink designs as well as meaningful demonstrations of the potential impact of the concepts. Overall, this is an interesting work that is appropriate for Nature Communications with moderate changes.

Suggested improvements:

Major suggestions:

1. The manuscript mentions the adhesion of the PaLMP film as a major advantage of this approach. However, it would be critical to demonstrate the adhesion of the process to other components. Figure S17 shows the attachment of conventional electronic components to the PaLMP using conductive epoxy. Can these structures be stretched (similar to Figure S6 in [Valentine et al, Adv Mater, 29:1703817, 2017])? Is it possible to use MGP to directly make contact to conventional electronic components (ie. start with a component on a substrate and print a PaLMP along the substrate and onto the component)?
2. What is the tensile modulus of the PaLMP? 1 MDa PSS can have a very high modulus.
3. "The LM+PSS solution had a spectrum representative of the combination of LM and PSS spectra, indicating that the PSS and LM are well-mixed in the solution."

It's unclear how the UV-vis data has any relationship to the mixing state. The interaction between the polymer and the LM can be investigated using approaches such as XPS [<https://pubs.acs.org/doi/10.1021/acsnano.1c05762?ref=PDF>].

Minor suggestions:

1. "Similar to bulk LM, printed PaLMP film shows negligible resistance variation under strain"

Bulk LM shows no resistivity change, but geometric resistance change (~8X increase at 200% strain). The results in this work are very different from what is expected from bulk LM [].

2. The rheology of the PaLMP ink is only reported for one concentration of LM. As the concentration of LM is increased, at what composition does the ink start to become non-Newtonian?

Writing suggestions:

3. "Liquid metal (LM) is being regarded as the most feasible material for soft electronics owing to its distinct combination of high conductivity [...]"

This is too subjective; there is no way to confirm that LM is being regarded as the "most feasible material".

4. "only the film printed with an AA-containing ink on a heated substrate was intrinsically conductive, with a resistivity of 6.66×10^{-7} ohm*m."

Other papers report this as a conductivity value rather than a resistivity value (eg. [Liu et al, Nat Mater, 20:851, 2021][Veerpandian et al, Nat Mater, 20:533, 2021]). It would be beneficial to report in the same way to help with comparisons.

5. Figure S3, S6, S17: The figures refer to SeLMP. Should this be PaLMP?

6. "These properties are realized through precise rheological control of designed PaLMP ink containing polyelectrolyte and acid, which are critically important to achieve stretchability and intrinsic conductivity."

It's unclear how the rheology determines the stretchability and conductivity. It seems like the rheology primarily impacts the printability.

Clarity and context: The scope of this work within the context of previous work has been presented clearly.

Reviewer #2 (Remarks to the Author):

In this paper, the author introduces liquid metal printing technique which can be electrically activated without any additional process. The author functionalizes the liquid metal particle with PSS and acetic acid using sonication for chemically stable and close packed structure under printing and stretching. It is interesting in that it can be printed and used as a conductive line without additional electrical activation. I recommend publication of this work, but additional explanations are required before publication.

1. In the description of clogging, does droplet mean lumps caused by liquid metal particle coalescence? Then, in figure 2G, coalescence is needed to increase the packing density of the printed particle. Is there a trade-off relationship between packing density and clogging? If so, what parameters should be controlled?

2. The author claims that PaLMP is patternable down to $50\mu\text{m}$. If narrow-level printing is an advantage, it would be good to show the conductivity by width of the line and high conductivity even at narrow width.

3. In figure 3 B and D, the author claims that meniscus-guided printing of liquid metal makes the printed line conductive by making conductive path through chemical sintering. The author should show the image of bottom view image of printed line as well as the cross-section and top.

4. Is the conductive path connected only through the sintered bottom? That is, is the top insulating and only the bottom conductive? If the resistance is issued only through the bottom, the side view of Figure 3D shows that the thickness of the sintering layer is very thin. What is the reason for the difference in resistance change according to the thickness during strain?

5. In Supplementary figure 3, PaLMP does not merge even when put in HCL and is chemically stable. In figure 7 D, why is it reduced when put in 5% AA solvent?

6. Here, the conductivity of the printed line is induced by chemical annealing. Therefore, the term "intrinsic" is not appropriate to this work.

7. Does the chemical annealing process depend on the particle size? Is it possible to create a conductive line using liquid metal nanoparticles?

Minor comments:

1. The authors mention that PaLMPs were prepared with 10 Vol.% AA in the results section. In contrast, the methods section states that PaLMPs were prepared with 5 Vol.% AA. Which is correct?
2. How can PaLMPs unstable (recycled) with weak acid (5 Vol. % AA) and stable with strong acid (HCl)?
3. The author used 20, 22, 25, 27-gauge needles for printing. Typically, the inner diameters of 20, 22, 25, and 27 gauge are 600, 400, 260, and 200 μm , respectively. Making lines with a resolution of 50 μm using a needle with a diameter of 200 μm is not reliable for large area printing.
4. The author mentioned that the diameter of the needle is 50 to 100 μm . That is, the needle size should be 34 gauge and 32 gauge, respectively. The authors need to recheck.

Reviewer #3 (Remarks to the Author):

This paper reports a very interesting research work, developing meniscus-guided printing of semi-solid-state polyelectrolyte-attached liquid metal microgranular-particles to pattern highly stable, ultra-stretchable, and intrinsically conductive electrodes. Overall, the key research findings and scientific results are clear and comprehensive.

However, some of the areas in the paper still need additional clarification to improve the manuscript's quality.

Major comments;

1. Liquid metal can be oxidized with electrical conductivity changes, which will be a major concern for applying sensors and electronics. However, there is no supporting information showing the printed materials' electrical stability and consistent conductivity over multiple days.
2. The printed electrodes need direct contact with the human skin to sense EMG and ECG data, which requires the validation of the material's biocompatibility/cytotoxicity. Therefore, please provide such information and data.
3. In addition, there is no photo showing the exact experimental setup to measure ECG and EMG data. It's unclear how the printed electrodes are connected to a circuit for data acquisition. Please add relevant photos and details of the information of the circuit and system.

Response to Reviewers' Comments

Title: Rapid meniscus-guided printing of stable semi-solid-state liquid metal microgranular-particle for soft electronics

Authors: Gun-Hee Lee†, Ye Rim Lee†, Hanul Kim, Do A Kwon, Hyeonji Kim, Siyoung Q. Choi, Jae-Woong Jeong*, and Steve Park*

Research Article No.: NCOMMS-21-48569

We thank the Reviewers for the careful consideration of our manuscript and the suggested demonstration, clarifications, and analyses that further strengthen our work. Following is a summary of our revision made according to the Reviewers' comments.

The Reviewer's comments are in **bold** and revised texts are **highlighted**.

Reviewer #1

Key Results: The manuscript describes the development of a liquid metal based ink that is suitable for meniscus-guided printing. The work includes control experiments to elucidate the effect of different ink components on the strain-dependent conductivity of the materials. A key part of the work is time-dependent pH as the aqueous solvent evaporates, which enables chemical dissolution of the oxide layer between microparticles, improving the conductivity. The ink is printed from an aqueous solvent, which is consistent with green processing concepts.

Validity: The manuscript includes some robust and well-supported conclusions, while a few of the conclusions are not well supported. These are discussed in the section on suggested improvements.

Significance: The work addresses several challenges with previous liquid metal printing methods that include limited print resolution and challenges with adhesion. Demonstrations include reprocessability of the starting materials through solvent dissolution, and piezoresistivity coefficients that depend on the thickness of the electrode. Consequently, this work includes significant technical insights into ink designs as well as meaningful demonstrations of the potential impact of the concepts. Overall, this is an interesting work that is appropriate for Nature Communications with moderate changes.

Answer. We appreciate the Reviewer for the helpful comment regarding our work. We have revised our work according to the Reviewer's comments. Point-by-point responses are attached below:

Suggested improvements:

Major suggestions:

1. The manuscript mentions the adhesion of the PaLMP film as a major advantage of this approach. However, it would be critical to demonstrate the adhesion of the process to other components. Figure S17 shows the attachment of conventional electronic components to the PaLMP using conductive epoxy. Can these structures be stretched (similar to Figure S6 in [Valentine et al, Adv Mater, 29:1703817, 2017])? Is it possible to use MGP to directly make contact to conventional electronic components (ie. start with a component on a substrate and print a PaLMP along the substrate and onto the component)?

Answer. We appreciate the Reviewer for the comment regarding stretchability. By using conductive epoxy, integrated electrical circuit can be stretched; however, its stretchability is limited to about 50% due to the rigidity of epoxy. As Reviewer suggested, we have integrated PaLMP with the conventional electronic components without epoxy. This configuration can easily be stretched and shows stable operation under 200% strain. We have added this test data in the Supplementary Information.

Revised Main Text

Repeated strain testing shown in **Fig. 5f** validates mechanical robustness and reliability of the multilayered interconnects, which are essential for high-density soft electronics. Also, PaLMP shows stable electrical conductivity without any encapsulation over several weeks (**Supplementary Fig. 21**). Finally, to confirm its integration capability with conventional electronics, we demonstrated the fabrication of a deformable micro-LED display (**Supplementary Fig. 22** and **Fig. 1g**) by connecting conventional electronic components with PaLMP devices using silver paste and conductive epoxy. Furthermore, electronic components can be directly integrated with PaLMP without conductive paste (**Supplementary Fig. 23**).

2. What is the tensile modulus of the PaLMP? 1 MDa PSS can have a very high modulus.

Answer. We thank the Reviewer for the comment regarding the mechanical property of PaLMP. Since the PaLMP film is not freely stand-alone, we have conducted experiment by coating it onto a thin PDMS film (100 μm). We have confirmed that the mechanical property of PDMS substrate before and after coating the PaLMP film are similar. PSS may have high modulus, however, it only covers the surface of the LM particles; hence, we expect it to have a negligible effect.

We have added the force as a function of strain in samples with and without PaLMP film in the Supplementary Information.

Added Figure and caption in Supplementary Information

Supplementary Fig. 17 | Force as a function of the strain of PDMS coated with PaLMP.

Revised Main Text

This confirms that PSS not only encapsulates the liquid metal particles to prevent GaIn from being extruded out, but also bridges the particles together^{34,35} to render the film stretchable. (see **Supplementary Fig. 17** for the force as a function of the strain of PDMS coated with PaLMP).

3. “The LM+PSS solution had a spectrum representative of the combination of LM and PSS spectra, indicating that the PSS and LM are well-mixed in the solution.”

It’s unclear how the UV-vis data has any relationship to the mixing state. The interaction between the polymer and the LM can be investigated using approaches such as XPS [<https://pubs.acs.org/doi/10.1021/acsnano.1c05762?ref=PDF>].

Answer. We thank the Reviewer for the comment regarding the data analysis. To confirm the attachment of polymer on the surface of LM particles, we have added EDS data in Supplementary Information. We have revised the Main Text according to the Reviewer’s comment. We have also deleted the abovementioned statement suggesting well-mixed solution with UV-spectra data.

Added Figure and caption in Supplementary Information

Supplementary Fig. 3 | Energy dispersive X-ray spectroscopy (EDS) data of PaLMP.

Carbon, which is a main element of PSS, is dispersed on the surface of LM particles.

Revised Main Text

Fig. 2b shows the UV-vis spectra of various solutions (all solutions had the same solvent consisting of water and AA): solvent only, LM, PSS, and LM+PSS. The LM+PSS solution had a spectrum representative of the combination of LM and PSS spectra. ~~indicating that the PSS and LM are well-mixed in the solution.~~ **Supplementary Fig. 3** shows the Energy dispersive X-ray spectroscopy (EDS) mapping, indicating that PSS are well-attached on the surface of LM particles.

Minor suggestions:

1. “Similar to bulk LM, printed PaLMP film shows negligible resistance variation under strain”

Bulk LM shows no resistivity change, but geometric resistance change (~8X increase at 200% strain). The results in this work are very different from what is expected from bulk LM.

Answer. We appreciate the Reviewer for the helpful comment. We have mistakenly added “similar to bulk LM” in the sentence above. Our film, contrary to bulk liquid metal film, shows negligible resistance change under strain, likely due to “positive piezo-conductivity.” We have revised our manuscript based on the Reviewer’s comment and added reference that explain the concept of positive piezo-conductivity.

Revised Main Text and added References.

Fig. 5a shows the resistance change of PaLMP printed on a VHB tape according to strain. Unlike bulk LM that shows resistance variance according to the change of geometry^{36,37}, printed PaLMP film shows negligible resistance variation under strain (e.g., an increase of 0.1 Ω at 200% strain) due to its piezo positive conductivity derived from particle-packed morphology^{38, 39}.

36. Leber, A. et al. Soft and stretchable liquid metal transmission lines as distributed probes of multimodal deformations. *Nat. Electron.* 3, 316-326, (2020).
37. Zhu, S. et al. Ultrastretchable Fibers with Metallic Conductivity Using a Liquid Metal Alloy Core. *Adv. Funct. Mater.* 23, 2308-2314, (2013).
38. Zheng, L. J. et al. Conductance-stable liquid metal sheath-core microfibers for stretchy smart fabrics and self-powered sensing. *Sci. Adv.* 7, (2021).
39. Yun, G. L. et al. Liquid metal-filled magnetorheological elastomer with positive piezoconductivity. *Nat. Commun.* 10, (2019).

2. The rheology of the PaLMP ink is only reported for one concentration of LM. As the concentration of LM is increased, at what composition does the ink start to become non-Newtonian?

Answer. We appreciate the Reviewer for the helpful comment regarding rheological property. PaLMPs become non-Newtonian at 30 v/v% showing shear thinning and yield stress in the flow curve. We have added flow curve measurement of PaLMP inks at various LM contents.

Revised Supplementary Information

Supplementary Fig. 8 | Rheological flow curve of PaLMP inks. Filled circles indicate flow curve of 10 v/v% PaLMP and gray line is its linear fit. Open data points are PaLMP with higher LM volume fractions (30 and 50 v/v %). The red line is the flow curve of acetic acid solution.

Rheological properties of PaLMPs were controlled by varying LM volume fraction. Shear rate-stress flow curve is measured with the strain stress control rheometer (MCR 302, Anton Paar). Shear rate range is from 0.1 to 102 1/s with logarithmic ramp up condition and the test time between each point was set as 60s to reach steady state. The torque limit was 0.1 $\mu\text{N}\cdot\text{m}$ which is sufficiently lower than the measuring range. Parallel plate (PP25) is used. For 10 v/v% sample, cylinder type cell (DG26.7/T200/SS) is used for the sufficient torque.

In Supplementary Fig. 8, the PaLMP sample with 10 v/v% LM shows Newtonian behavior with viscosity 14 mPa·s as the slope is very close to the unity in the logarithmic graph. When the LM volume fraction is increased to 30%, the viscosity at low shear rate increase more than 10 times. At 50 v/v%, the yield stress and shear thinning behavior appear to be stronger as typical characteristics of non-Newtonian fluids. For delicate printing of PaLMP at high resolution, 10 v/v% PaLMP with low viscosity is used throughout the demonstration.

Writing suggestions:

3. “Liquid metal (LM) is being regarded as the most feasible material for soft electronics owing to its distinct combination of high conductivity [...]”

This is too subjective; there is no way to confirm that LM is being regarded as the “most feasible material”.

Answer. We thank the Reviewer for the helpful comment. We have revised our manuscript based on the Reviewer’s comment.

Revised Main Text

Liquid metal (LM) is being regarded as a promising material for soft electronics owing to its distinct combination of high conductivity comparable to that of metals and exceptional deformability derived from its liquid state.

4. “only the film printed with an AA-containing ink on a heated substrate was intrinsically conductive, with a resistivity of 6.66×10^{-7} ohm*m.”

Other papers report this as a conductivity value rather than a resistivity value (eg. [Liu et al, Nat Mater, 20:851, 2021][Veerpandian et al, Nat Mater, 20:533, 2021]). It would be beneficial to report in the same way to help with comparisons.

Answer. We thank the Reviewer for the helpful comment. We have converted resistivity to conductivity (1.5×10^6 S/m) based on the Reviewer’s comment.

Revised Main Text

We have furthermore observed that only the film printed with an AA-containing ink on a heated substrate was intrinsically conductive, with a conductivity of 1.5×10^6 S/m.

5. Figure S3, S6, S17: The figures refer to SeLMP. Should this be PaLMP?

Answer. We appreciate the Reviewer for the careful review. We have revised our manuscript with the right terminology and have carefully reviewed our work to avoid any further mistakes.

Revised Supplementary Information.

Supplementary Fig. 4| Chemical stability of ink according to the presence and absence of PSS. a, Photograph of reduced LMP into bulk LM when HCl was added to the solution without PSS. **b,** Photograph of the PaLMP ink after the addition of HCl.

To test the stability of ink, 0.2 ml of HCl (37%) is added to 4 ml ink.

6. “These properties are realized through precise rheological control of designed PaLMP ink containing polyelectrolyte and acid, which are critically important to achieve stretchability and intrinsic conductivity.”

It’s unclear how the rheology determines the stretchability and conductivity. It seems like the rheology primarily impacts the printability.

Answer. We thank the Reviewer for the helpful comment. We understand that this is a confusing statement and we have deleted this sentence to make conclusion more clear to the reader.

Revised Supplementary Information.

Here, we developed such a technique based on MGP of PaLMP. PaLMP patterned with MGP is not only highly customizable in terms of design, but also stretchable, mechanically stable, and capable of high-resolution and multilayer patterning on various substrates without any additional process. ~~These properties are realized through precise rheological control of designed PaLMP ink containing polyelectrolyte and acid, which are critically important to achieve stretchability and intrinsic conductivity.~~

Reviewer #2

In this paper, the author introduces liquid metal printing technique which can be electrically activated without any additional process. The author functionalizes the liquid metal particle with PSS and acetic acid using sonication for chemically stable and close packed structure under printing and stretching. It is interesting in that it can be printed and used as a conductive line without additional electrical activation. I recommend publication of this work, but additional explanations are required before publication.

Answer. We appreciate the Reviewer for the comments regarding our work. We have revised our work according to the Reviewer's comment to improve the clarity of our work. Detail responses are attached below:

1. In the description of clogging, does droplet mean lumps caused by liquid metal particle coalescence? Then, in figure 2G, coalescence is needed to increase the packing density of the printed particle. Is there a trade-off relationship between packing density and clogging? If so, what parameters should be controlled?

Answer. We appreciate the Reviewer for the helpful comment. As the reviewer mentioned, the main reason for clogging in the case of bare liquid metal particle is their coalescence (i.e. breakage of oxide shell and complete merging) during printing due to its inability to sustain the applied shear force during extrusion. On the other hand, in the case of PaLMP, there are no coalescence during printing and it maintain its droplet assembled thin-film morphology. Therefore, we concluded that the main factor for reliable printing without clogging is stability of the particles that prevent the rupture and coalescence during printing.

We have added a SEM image of bare LM particles after clogging, which shows significant amount of rupture and coalescence. Clogging and packing density are unassociated effects. Clogging only occurs with bare LM particles; whereas, packing density is affected by the presence/absence of AA in the PaLMP solution.

Revised Figure and caption in Supplementary Information

Supplementary Fig. 5| SEM image of clogged bare LMP.

Due to instability, bare LMPs ruptured and coalesced during printing, which results in the clogging of the nozzle.

Revised Main Text

The nozzle extruding LMP ink often clogged during printing (**Supplementary Movie 1**); on the other hand, the PaLMP ink showed reliable printing without clogging (**Supplementary Movie 2**). Clogging can be explained by the coalescence of the LMP during printing due to its instability (**Supplementary Fig. 5**).

2. The author claims that PaLMP is patternable down to 50 μm . If narrow-level printing is an advantage, it would be good to show the conductivity by width of the line and high conductivity even at narrow width.

Answer. We appreciate the Reviewer for the helpful comment. We have added conductivity data according to the line width in the Supplementary Information. As printed line width decreased, conductivity decreased likely due to the diminishing of conductive pathways. However, printed PaLMP lines showed decent conductivity over 1×10^6 S/m

Revised Figure and caption in Supplementary Information

Supplementary Fig. 12| Printed PaLMP with different line width. a, OM image of printed PaLMP lines with different line width. b, Photograph of printing needles with different diameter. c, Conductivity of printed PaLMP line with different line width.

3. In figure 3 B and D, the author claims that meniscus-guided printing of liquid metal makes the printed line conductive by making conductive path through chemical sintering. The author should show the image of bottom view image of printed line as well as the cross-section and top.

Answer. We thank the Reviewer for the helpful comment. We have added a bottom image of printed line by printing PaLMP on a cover glass.

Revised Figure and caption in Supplementary Information

Supplementary Fig. 14| Top and bottom OM image of printed PaLMP. a, Top view. b, Bottom view.

The top layer of PaLMP film maintains its particle-packed morphology and the bottom layer of PaLMP film shows the sintered grain.

Revised Main Text

Thus, the annealing of PaLMPs in an acidic condition partially removes of the oxide shell, which extrudes out the gallium. We hypothesize that the removal and reformation of the oxide layer at the interface strengthens the adhesion of PaLMP film to the substrate^{21,33} (Supplementary Fig.14).

4. Is the conductive path connected only through the sintered bottom? That is, is the top insulating and only the bottom conductive? If the resistance is issued only through the bottom, the side view of Figure 3D shows that the thickness of the sintering layer is very thin. What is the reason for the difference in resistance change according to the thickness during strain?

Answer. We thank the Reviewer for the comment regarding conductivity. If only the bottommost layer was conductive, then the resistance change with respect to applied strain (gauge factor) should be similar for various electrode line thicknesses. However, since there is a difference in the gauge factor at different thicknesses, we can deduce that the conductive layer thickness is indeed varying. Also, XPS data shows the presence of Gallium at the top of the printed line, which suggests partial merging and electrical connection of PaLMP. The difference in conductivity at various positions along the thickness was not measurable. We presume that depending on the printed line thickness, the volume proportion that exhibits positive piezo conductivity varies, which is responsible for the difference in the gauge factor. However, the exact reason for the difference positive piezo conductivity at different line thicknesses requires further in-depth analysis, which we believe should be reserved for another separate study. We have added references and revised the Main Text to further explain the concept of positive piezo conductivity of the composite or particle-based liquid metal.

Revised Main Text and added References.

Fig. 5a shows the resistance change of PaLMP printed on a VHB tape according to strain. Unlike bulk LM that shows resistance variance according to the change of geometry^{36,37}, printed PaLMP film shows negligible resistance variation under strain (e.g., an increase of 0.1 Ω at 200% strain) due to its piezo positive conductivity derived from particle-packed morphology^{38, 39}

36. Leber, A. et al. Soft and stretchable liquid metal transmission lines as distributed probes of multimodal deformations. *Nat. Electron.* 3, 316-326, (2020).
37. Zhu, S. et al. Ultrastretchable Fibers with Metallic Conductivity Using a Liquid Metal Alloy Core. *Adv. Funct. Mater.* 23, 2308-2314, (2013).
38. Zheng, L. J. et al. Conductance-stable liquid metal sheath-core microfibers for stretchy smart fabrics and self-powered sensing. *Sci. Adv.* 7, (2021).
39. Yun, G. L. et al. Liquid metal-filled magnetorheological elastomer with positive piezoconductivity. *Nat. Commun.* 10, (2019).

5. In Supplementary figure 3, PaLMP does not merge even when put in HCL and is chemically stable. In figure 7 D, why is it reduced when put in 5% AA solvent?

Answer. We appreciate the reviewer for the helpful comment. In Figure 7d, 5% AA solvent is used to dissolve the biodegradable substrate. After the substrate is degraded, PaLMPs can be reduced to bulk LM by HCl. Here, “reduction by acid” means “reduction by HCl”. Here we used commercially purchased HCl (with a concentration of 37%). To avoid confusion we have added related information.

In Supplementary Figure 3, we added 0.2 ml of HCl to 4 ml of solution. Under this diluted acid condition, PaLMP does not get reduced due to its chemical stability; however, bare LMP gets reduced. We have revised our Supplementary Information by adding experiment information to avoid any confusion.

Revised Main Text

Different from many skin-attachable sensors based on conventional metals⁴⁹⁻⁵², this PaLMP-based ECG sensor can be fully disposed without waste as presented in Fig. 7d. Firstly, biodegradable substrate is fully degraded by immersing in the diluted weak acid solution (5 % AA). Subsequently, all of PaLMPs were reduced by the addition of HCl (37%) and merged into bulk LM droplets that are reusable. This demonstration shows the possibility of using PaLMP for construction of skin-compatible, deformable, and recyclable electronics, suggesting new opportunities for zero-waste medical devices.

Revised Supplementary Information.

Supplementary Fig. 4| Chemical stability of ink according to the presence and absence of PSS. a, Photograph of reduced LMP into bulk LM when HCl was added to the solution without PSS. **b,** Photograph of the PaLMP ink after the addition of HCl.

To measure the stability of ink, 0.2 ml of HCl (37%) is added to 4 ml ink.

6. Here, the conductivity of the printed line is induced by chemical annealing. Therefore, the term “intrinsic” is not appropriate to this work.

Answer. We appreciate the reviewer for the comment regarding the terminology using our work. We have changed ‘intrinsic conductivity’ to ‘initial conductivity’ and ‘intrinsically conductive’ to ‘initially conductive’.

7. Does the chemical annealing process depend on the particle size? Is it possible to create a conductive line using liquid metal nanoparticles?

Answer. We appreciate the reviewer for the comment regarding the modification of liquid metal particles. Liquid metal particle size decreases with increasing tip sonication time as shown in the Supplementary Figure 1 below. The size of our particles does not get reduced down to nanoscale; the size saturates at about a few micrometers likely due to the instability of the particles below this size.

Supplementary Figure 1. Radius of PaLMP according to sonication time.

Minor comments:

1. The authors mention that PaLMs were prepared with 10 Vol.% AA in the results section. In contrast, the methods section states that PaLMs were prepared with 5 Vol.% AA. Which is correct?

Answer. We appreciate the Reviewer for the careful review of our work. We have found our mistake and revised the method section. We used 10 vol.% AA for the printing. We have also carefully reviewed the paper to avoid any further mistakes.

Revised method section.

Preparation of PaLMP ink for MGP. Bulk LM (1.25 g) and 0.07 g (7×10^{-8} mol) of PSS were added in diluted acetic acid (10 vol% in DI water, 2 ml). This compound solution was tip sonicated (VC 505, Sonics & Materials, 3 mm microtip) at 500 W and 20 kHz for 30 minutes.

2. How can PaLMs unstable (recycled) with weak acid (5 Vol. % AA) and stable with strong acid (HCl)?

Answer. We appreciate the Reviewer for the careful review of our work. PaLMP is not reduced by a weak acid. In Figure 7d, 5% AA solvent is used to dissolve the biodegradable substrate. After the substrate is degraded, PaLMs can be reduced to bulk LM by high concentration HCl. “Reduction by acid” in Fig.7d means “reduction by HCl”. Here, we used commercially purchased HCl (with a concentration of 37%). We have revised the Main Text.

In Supplementary Figure 3, we added 0.2 ml of HCl (37%) to 4 ml of solution. Under this diluted acid condition, PaLMP does not get reduced due to its chemical stability; however, bare LMP gets reduced. We have revised our Supplementary Information by adding experiment information to avoid any confusion.

Revised Figure and Main Text

Different from many skin-attachable sensors based on conventional metals⁴⁹⁻⁵², this PaLMP-based ECG sensor can be fully disposed without waste as presented in Fig. 7d. Firstly, biodegradable substrate is fully degraded by immersing in the diluted weak acid solution (5 vol.% AA). Subsequently, all of PaLMs were reduced by the addition of HCl (37%) and merged into bulk LM droplets that are reusable. This demonstration shows the possibility of using PaLMP for construction of skin-compatible, deformable, and recyclable electronics, suggesting new opportunities for zero-waste medical devices.

Revised Supplementary Information.

Supplementary Fig. 4| Chemical stability of ink according to the presence and absence of PSS. a, Photograph of reduced LMP into bulk LM when HCl was added to the solution without PSS. **b,** Photograph of the PaLMP ink after the addition of HCl.

To test the stability of ink, 0.2 ml of HCl (37%) is added to 4 ml ink.

3. The author used 20, 22, 25, 27-gauge needles for printing. Typically, the inner diameters of 20, 22, 25, and 27 gauge are 600, 400, 260, and 200 μm , respectively. Making lines with a resolution of 50 μm using a needle with a diameter of 200 μm is not reliable for large area printing.

Answer. We appreciate the Reviewer for the careful review of our work. We have found that we omitted the picture of small size nozzles. We have added a picture with the small size needles in the Supplementary Information.

Supplementary Fig. 12 | Printed PaLMP with different line width. **a**, OM image of printed PaLMP lines with different line width. **b**, Photograph of printing needles with different diameter. **c**, Conductivity of printed PaLMP line with different line width.

4. The author mentioned that the diameter of the needle is 50 to 100 μm . That is, the needle size should be 34 gauge and 32 gauge, respectively. The authors need to recheck.

Answer. We appreciate the Reviewer for the careful review of our work. We have found our mistake and added a picture of 34 and 32 gauge needles in the Supplementary Information.

Reviewer #3

This paper reports a very interesting research work, developing meniscus-guided printing of semi-solid-state polyelectrolyte-attached liquid metal microgranular-particles to pattern highly stable, ultra-stretchable, and intrinsically conductive electrodes. Overall, the key research findings and scientific results are clear and comprehensive.

However, some of the areas in the paper still need additional clarification to improve the manuscript's quality.

Answer. We appreciate the Reviewer for the acknowledgment of our work. We have revised our manuscript with the additional information to clarify the manuscript.

Major comments;

1. Liquid metal can be oxidized with electrical conductivity changes, which will be a major concern for applying sensors and electronics. However, there is no supporting information showing the printed materials' electrical stability and consistent conductivity over multiple days.

Answer. We appreciate the Reviewer for the helpful comment regarding electrical conductivity. We have conducted an additional experiment to measure the resistance of the line for 30 days after printing. Although the resistance of the printed line slightly increased after a few days, it saturated and showed good electrical conductivity after 30 days.

Added Figure and caption in Supplementary Information

Supplementary Fig. 21| Resistance variation of printed PaLMP line for 30 days.

Revised Main Text

Repeated strain testing shown in **Fig. 5f** validates mechanical robustness and reliability of the multilayered interconnects, which are essential for high-density soft electronics. **Also, PaLMP shows stable electrical conductivity without any encapsulation over several weeks (Supplementary Fig. 21).**

2. The printed electrodes need direct contact with the human skin to sense EMG and ECG data, which requires the validation of the material's biocompatibility/cytotoxicity. Therefore, please provide such information and data.

Answer. We appreciate the Reviewer for the helpful comment. To verify the biocompatibility of PaLMP, we have conducted in vitro cytotoxicity test with 3T3 cells. The following figures present both bright view and live/dead staining images for the observation of cell morphology and viability of cells, which showed high cell viability after incubation with our material (PaLMP) for 24 hrs indicating the biocompatibility of PaLMP.

We have added biocompatibility data in the Supplementary Information.

Added Figure and caption in Supplementary Information

Supplementary Fig. 29| Biocompatibility of PaLMP film. a, Bright field and fluorescent images of 3T3 cells on SBS film b, Bright field and fluorescent images of 3T3 cells on PaLMP coated SBS film.

The proliferation of 3T3 cells on PaLMP indicates its biocompatibility suitable for wearable application.

Revised Main Text

Here, the versatile printing-based manufacturability and high mechanical/chemical stability of PaLMP enable construction of deformable EMG sensors customized for desired applications, as presented in Fig. 6f. Another significant attribute of the PaLMP-based EMG electrode is that it is biocompatible (Supplementary Fig. 29) and does not leave residues on the skin after usage, making it highly suitable for skin-interfaced use.

3. In addition, there is no photo showing the exact experimental setup to measure ECG and EMG data. It's unclear how the printed electrodes are connected to a circuit for data acquisition. Please add relevant photos and details of the information of the circuit and system.

Answer. We appreciate the Reviewer for the helpful comment. To measure the ECG and EMG data, we have used commercialized Bluetooth EP monitoring sensing system (BioRadio, Great Lakes NeuroTech.). In the case of EMG monitoring, we used copper tape to connect the fabricated sensor and the system and used medical tape-based insulating layer to avoid noise. In the case of ECG monitoring, we used copper tape to connect the fabricated sensor and the system.

We have added a measurement setup in Supplementary Information and revised method section with additional information.

Added Figure and caption in Supplementary Information

Supplementary Fig. 30| Experimental setup for EMG measurement. a, Photograph of setup for monitoring EMG. b, Schematic illustration of PaLMP-based EMG sensor.

A commercialized Bluetooth sensing device (BioRadio, Great Lakes NeuroTech.) was used for acquiring physiological signals (EMG, ECG).

Revised Main Text

Fig. 6g shows EMG signals measured with a PaLMP-based sensor on a forearm, verifying its ability of high fidelity EMG sensing (setup for physiological signal measurement is presented in **Supplementary Fig. 30**).

Fabricated EMG and ECG sensors are connected with a commercialized Bluetooth EP signal sensing system (BioRadio, Great Lakes NeuroTech.).

REVIEWERS' COMMENTS

Reviewer #1 (Remarks to the Author):

The authors have addressed the comments and concerns of the reviewers, and I believe the manuscript is ready for publication.

Reviewer #2 (Remarks to the Author):

This reviewer thinks the authors properly addressed the comments, so recommends publication.

Reviewer #3 (Remarks to the Author):

The revised manuscript is in great shape with additional data and supporting information. Responses to the given comments are very satisfactory. Therefore, this manuscript should be accepted.

Response to Reviewers' Comments

Title: Rapid meniscus-guided printing of stable semi-solid-state liquid metal microgranular-particle for soft electronics

Authors: Gun-Hee Lee†, Ye Rim Lee†, Hanul Kim, Do A Kwon, Hyeonji Kim, Siyoung Q. Choi, Jae-Woong Jeong*, and Steve Park*

Research Article No.: NCOMMS-21-48569

We thank the Reviewers for the careful consideration of our manuscript and the suggested demonstration, clarifications, and analyses that further strengthen our work. Following is a summary of our revision made according to the Reviewers' comments.

The Reviewer's comments are in **bold** and revised texts are **highlighted**.

Reviewer #1

The authors have addressed the comments and concerns of the reviewers, and I believe the manuscript is ready for publication.

Answer. We appreciate the Reviewer for careful review of our work.

Reviewer #2

This reviewer thinks the authors properly addressed the comments, so recommends publication.

Answer. We appreciate the Reviewer for careful review of our work.

Reviewer #3

The revised manuscript is in great shape with additional data and supporting information. Responses to the given comments are very satisfactory. Therefore, this manuscript should be accepted.

Answer. We appreciate the Reviewer for careful review of our work.